# HPV Vaccination Intentions of Female Students in Chinese Universities: A Systematic Literature Review and Meta-Analysis

**DOI:** 10.3390/ijerph191610207

**Published:** 2022-08-17

**Authors:** Yiming Bai, Patrick Ip, Karen Chan, Hextan Ngan, Paul Yip

**Affiliations:** 1Department of Social Work and Social Administration, Faculty of Social Sciences, The University of Hong Kong, Hong Kong, China; 2Department of Paediatrics and Adolescent Medicine, The University of Hong Kong (HKU), Hong Kong, China; 3Department of Obstetrics and Gynaecology, School of Clinical Medicine, The University of Hong Kong, Hong Kong, China

**Keywords:** HPV vaccine, Chinese female university students, vaccination intention, meta-analysis

## Abstract

Objective: To systematically evaluate the human papillomavirus (HPV) vaccination intentions among female university students in China and establish a basis for improving HPV vaccination coverage. Methods: We searched CNKI, EBSCO, JSTOR, MESH or Emtree, Weipu Information Chinese Journal Service Platform, Wanfang Data, China Biomedical Literature Database, PubMed, Embase, Web for the Web of Science, and the Cochrane Library to identify peer-reviewed published research on intentions by female college students in China to receive the HPV vaccination. Results: A preliminary search of 408 papers resulted in the inclusion of 12 studies, all cross-sectional, of moderate or high quality, with a sample size of 12,600. The HPV vaccination intention rate among Chinese female university students was 16.67% (95% CI: 12.38% to 21.24%). The vaccination intention rates of medical students, non-medical students, and Tibetan students were 30.37% (95% CI: 28.80–34.12%), 15.53% (95% CI: 11.2–20.22%), and 14.12% (95 % CI: 10.59–18.04%), respectively. The vaccination intention rates of the participants with parental education of junior high school and below, high school, and bachelor’s degree and above were 15.36% (95 % CI: 11.59 to 17.54%), 17.18% (95 % CI: 12.33% to 19.61%), and 19.81% (95 % CI: 15.61% to 22.25%), respectively. The intention rates of vaccination among residents of first-tier, second-tier, and third-tier cities were 17.64% (95% CI: 12.76–21.63%), 15.39% (95% CI: 11.74–19.82%), and 13.87% (95% CI: 9.36–15.65%), respectively. The results of the meta-analysis were relatively stable with little publication bias. Conclusion: The intention rate of HPV vaccination among female university students in China is low and varies among different populations. There is a need to increase HPV vaccination promotion efforts to improve the intention of female university students to receive the vaccine.

## 1. Introduction

Cervical cancer is the fourth most common type of female cancer and has the fourth highest mortality rate [1]. There were approximately 57,000 new cases of cervical cancer and approximately 310,000 deaths worldwide in 2021. Cervical cancer is closely associated with human papillomavirus (HPV) infection. [2] HPV vaccination prevents over 70% of cervical cancers and is the most effective method of preventing acromegaly and infections-related diseases [3]. The highest risk group for HPV infection are women between the ages of 19 and 44 years [3]. The Global Advisory Committee on Vaccine Safety (GACVS) has evaluated the safety of HPV vaccines on multiple occasions, and concluded in 2017 that HPV vaccines have an extremely favorable safety profile [4]. The cumulative benefits of HPV vaccination far outweigh the risks, with the greatest benefit occurring when the entire course of HPV vaccination is completed prior to initiating sexual intercourse. The World Health Organization (WHO) recommends that the HPV vaccine be included in the national vaccine plan. Vaccinating female university students against HPV has been shown to be cost-effective [4].

At present, there are three types of HPV vaccines worldwide: bivalent, quadrivalent, and nine-valent vaccines. [5] The quadrivalent vaccine was launched in the United States in June 2006 [6]. It protects against HPV subtypes 6, 11, 16, and 18; in addition, it has been marketed in more than 100 countries worldwide. The bivalent vaccine, which was subsequently released, targets two high-risk HPV subtypes: 16 and 18. [7] The nine-valent vaccine was the last to be released; however, it has received the most public attention. It protects against five subtypes of HPV—31, 33, 45, 52, and 58—in addition to the four types of quadrivalent vaccine, which are closely related to cervical, vulvar, and vaginal cancers in women [8]. The HPV bivalent vaccine obtained the first marketing approval in China in July 2016. The quadrivalent vaccine was launched in China in November 2016, and the nine-valent HPV vaccine was launched in China on 28 April 2018. Thus, all the HPV vaccines are now available in China. The HPV vaccine is import-dependent and is in high demand in China. The availability of bivalent and quadrivalent vaccines is generally covered; however, the nine-valent vaccine is still in short supply. It is worth noting that China’s local HPV research and development has been initiated; in addition, eight institutions have been set up to work on HPV vaccine development, among which, Yunnan Watson’s nine-valent vaccine has been approved for clinical research. The launch of the domestic HPV vaccine will alleviate the problem of insufficient HPV vaccine supply. In China, the bivalent vaccine is available for women aged 9–45 years. The quadrivalent vaccine is available for ages 20–45, and the nine-valent vaccine is available for ages 16–26. The absence of a national immunization program and the age limit for vaccination determine the difficult environment for HPV vaccination in China; moreover, there is a large gap with developed regions. Many scholars have investigated the level of awareness and vaccination coverage of the HPV vaccine among Chinese women; they found that the awareness of the HPV vaccine is generally insufficient and the number of vaccinations is low in China [9]. The literature reported that the negative attitude and behaviors of Chinese female college students in the mainland, such as a lack of knowledge and low awareness about the mode of transmission, protection, and early diagnosis methods of HPV infection, the cost of the HPV vaccine, potential adverse effects, suspecting vaccine safety, and the negative news on all vaccines, prevented the generalization of the HPV vaccination [10]. The level of awareness and acceptance of HPV and the HPV vaccine in Hong Kong, Macao, and Taiwan is generally higher than that in the Mainland. Hong Kong female residents themselves have a high acceptance of the HPV vaccine; however, their acceptance of vaccination for their children is low. In addition, there is a low-vaccination rate in practice [11]. This systematic review included full-time female college students aged 17–24 years old, who constitute a large portion of the vulnerable population aged 17 to 44. The vaccination is more effective at younger ages under the condition of vaccination availability, and it is more likely to be efficient in controlling the infection among the risk group of students who are sexually active if they can receive an effective transmission intervention [12]. The HPV vaccine is a class II vaccine with the characteristics of a commodity in China. College students have a certain self-determination in terms of their financial situation, and their willingness to receive the HPV vaccination depends more on their own intention; however, it is also influenced by their parents’ opinion toward the vaccine, which directly affects the vaccination rate.

We were aware that many studies on HPV vaccination among female college students in China had been conducted; however, the research findings varied due to differences in sampling, sample sizes, research designs, population tendencies, and data capture methods. There were no multi-center, large-scale research publications.

This article systematically reviews and conducts a meta-analysis of the research literature to explore the HPV vaccination intentions of female college students in China. We make recommendations for increasing HPV vaccination rates based on the findings.

## 2. Methods

### 2.1. Inclusion and Exclusion Criteria

We included original cross-sectional and cohort studies that reported on the HPV vaccination status of female college students in China (aged 17 to 24 years). Papers should describe characteristics of students who received or intended to receive the HPV vaccination. Data on intention should be provided as the relative risks of vaccination intention (including measures of variability), and factors associated with vaccination decisions. The outcome was the intention to be vaccinated for cervical cancer or to receive an HPV vaccination.

We excluded literature with incomplete data, literature not in Chinese or English, summaries of reviews, and conference abstracts. If there were two or more publications from the same study, complete data and detailed reports were selected across the studies.

### 2.2. Search Strategy

The literature search used a combination of subject terms and free words, Boolean logic operators, and word truncation symbols, in CNKI, EBSCO, JSTOR, Weipu Information Chinese Journal Service Platform, Wanfang Data, China Biomedical Literature Database, MESH, Emtree, PubMed, Embase, Web for the Web of Science, and the Cochrane Library; we searched the literature published before 1 October 2021. We did not restrict the literature publication status during the search process. We also read the references of the included literature to supplement the search.

Search terms: Chinese search terms: (“Cervical cancer vaccine” OR “HPV vaccine” OR “HPV vaccine” OR “6-valent HPV vaccine” OR “9-valent HPV vaccine”) AND “vaccination” AND (“willingness” OR “attitude” OR “intention”); English search terms: (“HPV” OR “HPV-6” OR “HPV-9” OR “papillomavirus virus vaccine”) AND (“attitude *” OR “opinion *” OR “intention *”).

### 2.3. Literature Screening and Data Extraction

One researcher reviewed the titles and abstracts of the literature; then, the researcher read the full text of potentially-relevant articles, and determined the final inclusion decision based on inclusion and exclusion criteria. Baseline information was extracted about the first author, year of publication, study population, sample size, and parental education level.

### 2.4. Study Quality Assessment

The methodological quality of the included studies was evaluated using the six item Downs and Black grading method [6]. Research with high methodological quality scored 5 or 6 points, medium-quality research scored 3–4 points, and low-quality research scored 3 or fewer points.

### 2.5. Data/Statistical Analysis

R software was used to conduct meta-analysis. The Q test (*p*-value) and I2 test the heterogeneity of study findings. If *p* > 0.10 and I2 ≤ 50%, the fixed-effects model should be adopted; if *p* < 0.10 and I2 > 50%, the random-effects model should be adopted. Subgroup analysis was carried out where possible, for students’ major areas of study, parents’ educational level, and city type. Sensitivity analysis evaluated the stability of the research results; after the exclusion of a particular low-quality literature, the merged effect size was re-estimated and compared with the meta results before the exclusion to explore the extent of the influence of the particular study on the merged effect size and the robustness of the results. It indicated that the sensitivity was low and the results were robust if the results did not change significantly after exclusion. Begger’s test was used to judge the publication bias. The level of significance was α = 0.05.

### 2.6. Results

The searches identified 408 potentially-relevant documents. After deduplication and screening, 12 studies were included [8,9,10,11,12,13,14,15,16,17,18,19,20]. The PRISMA flowchart outlining the screening process is shown in Figure 1.

There were three high-quality papers and nine medium-quality papers. The sample size was 12,600 young women, and descriptors of the included literature are reported in Table 1.

## 3. Meta-Analysis Results

The included studies showed significant heterogeneity (I^2^ = 95.7%, *p* < 0.001); thus, a random-effects model was used for meta-analysis. The overall weighted intention rate of HPV vaccination among Chinese female college students was less than 16.67% (95% CI: 12.38~21.24%), as shown in Figure 2.

### 3.1. Subgroup Analysis

The subgroup analysis identified important subgroup differences. Medical students had the highest vaccination intention of 30.37% (95% CI:28.80~34.12%). The intention rate of vaccination increased with increases in parental education level: for girls with parents who had an undergraduate education or above, it was 19.81% (95% CI:15.61~22.25%); for girls whose parents had a high school education, 17.18% (95% CI:12.33~19.61%); and for girls whose parents had only up to a junior high school education, 15.36% (95% CI:11.59~17.54%). However, the non-medical college students’ intention rate of 15.53% (95% CI:11.2~20.22%) had almost half the medical student’s intention rate. The intention rate for HPV vaccination among female Tibetan college students is 14.12% (95% CI:10.59~18.04%), which was marginally lower than medical students. The intention rates of HPV vaccination among female college students in first-tier, second-tier, and third-tier cities were 17.64% (95% CI:12.76~21.63%), 15.39% (95% CI:11.74~19.82%), and 13.87% (95% CI:9.36%~15.65%), respectively. The higher the city rated, the higher was the intention rate to vaccinate; see Table 2.

### 3.2. Sensitivity Analysis and Publication Bias

After excluding the low-quality papers, the effect size was merged again; the results did not change significantly, as shown in Table 3. Begger’s test showed a minimum publication bias (*Z* = 0.08, *p* = 0.9379).

## 4. Discussion

This paper provides the first overview of female Chinese college students’ intentions regarding the HPV vaccine. There were differences in the intention with respect to HPV vaccination among specialization, ethnic groups, parental education levels, and city level. The intention rate of medical students to vaccinate is much higher than any other group, which may be related to their better knowledge of the vaccination’s benefit. The inoculation intention rate in girls with parents with a higher education was much higher than that of girls whose parents had a lower education. Residents in first-tier and second-tier cities are more willing to be vaccinated possibly because they have more ways to obtain health information, better access to the vaccine, and are more likely to be able to afford it [19,20,21,22]. Higher acceptance levels of the HPV vaccine may reflect better promotion by high city-level communities than lower status cities, which may influence young women’s knowledge and awareness. [23] First-tier cities such as Beijing, Shanghai, and Shenzhen have started free vaccination programs for adolescents; in addition, some other regions are advocating the inclusion of HPV vaccines in health insurance reimbursements to increase residents’ willingness to receive vaccination [24]. A high level of uptake (43%) has been reported in countries where the HPV vaccine is free-of-charge under a government-mandated immunization schedule; for example, in the UK and Australia, compared to 27% before the free vaccination program was implemented [25,26]. In Hong Kong, the intention rate is higher than the other regions in China. In particular, since 2019, the Hong Kong government has provided free HPV vaccination to girls in primary school. [27] The first vaccination rate has reached 41.4% among them; moreover, the two-vaccination completion rate has reached 29.2%, compared with 12% before implementing the free program [28]. The cost of the vaccination in the mainland, ranging from several hundred to thousands of RMB for each dose, can be substantial [29,30]. The cost of the HPV nine-valent vaccine in Beijing, for example, is nearly 4900 RMB for the completion of three doses [31]. (The Renminbi (RMB) is the official name of China’s currency.) In China, the hurdles to HPV vaccination include worries about vaccine safety and efficacy, side effect concerns, a lack of or inconsistent information regarding HPV and the HPV vaccine, and the cost of the HPV vaccine.

This review included 12 cross-sectional and cohort studies of 11,558 college students across 12 cities. In order to improve vaccination rates, it is necessary to actively increase the willingness of young women at university, aged 18–22, to receive HPV vaccination. Based on our findings, it is pertinent to introduce courses on the prevention and treatment of common diseases such as HPV and vaccines into undergraduate education of non-medical students, including ethnic Tibetan students, in order to improve the vaccination intentions of Chinese female college students. Additionally, medical students might be recruited as peer leaders on university campuses in order to positively influence the intentions of other female college students to receive HPV vaccinations [32,33,34,35]. To address female college students’ concerns about vaccination safety and cost, as well as to strengthen HPV vaccination education among college students, it is recommended that the government provide appropriate subsidies to reduce the burden of costs of vaccination for high-risk groups, and improve accessibility. Prior to HPV vaccination, female college students should receive standard, comprehensive, scientific, objective, and unbiased vaccination advice, including adequate information about the risks of cervical cancer and HPV infection-related diseases, as well as unbiased information about HPV vaccine coverage; in addition, they should be advised that additional cervical cancer prevention and treatment measures should not be neglected following the vaccination.

### 4.1. Limitations

This study has the following limitations: (1) we were unable to examine the difference between studies on full HPV vaccination and those only receiving first dose vaccines. The intention to undertake and complete the vaccination course may be very different; (2) due to the limitations of the individual studies, there was a large heterogeneity in the rates between studies. However, the subgroup analysis did not find the source of heterogeneity; (3) it was not possible to compare subgroups directly; (4) there are many types of HPV vaccines, and there was potentially no difference in intention to vaccinate using particular types of HPV vaccine. Dissecting this was not possible; and (5) whether the HPV vaccine was freely available in each study is unknown; thus, it was impossible to compare differences in vaccination intention between free vaccinations and self-funded vaccinations.

### 4.2. Conclusions

In summary, Chinese female college students have a relatively low intention to vaccinate for HPV compared with other countries. In Australia, HPV vaccination is ahead of other countries; Australia may be the first country to be free of the disease. In 2007, the Australian federal government began providing the vaccine free of charge to girls aged 12–13 years; it then expanded the program to boys in 2013. In 2016, 78.6% of 15-year-old girls and 72.9% of 15-year-old boys had been vaccinated. As a result, the prevalence of HPV among women aged 18–24 years decreased from 22.7% to 1.1% between 2005 and 2015 [36,37,38,39].

Cervical cancer is a preventable disease, which if not arrested has significant personal and medical costs associated with it. It is recommended that the HPV vaccine be included in the Chinese Government immunization plans and medical insurance reimbursement schemes as soon as possible; this would reduce the economic and social burdens of contracting the disease, and remove the barrier of affordability. Schools and community hospitals are encouraged to carry out HPV vaccination and related disease knowledge promotion activities regularly in order to raise parents’ and young women’s awareness of the value of HPV vaccines; this would thereby increase their intention to be vaccinated.

## Figures and Tables

**Figure 1 ijerph-19-10207-f001:**
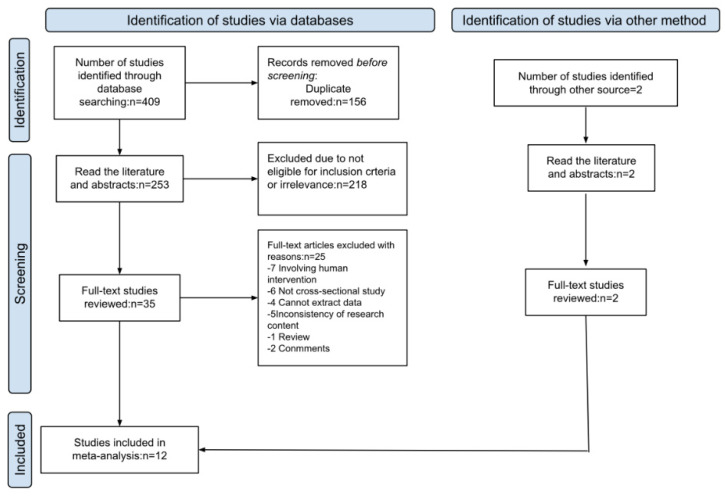
Flow diagram representing the process of identifying and including studies in the meta-analysis.

**Figure 2 ijerph-19-10207-f002:**
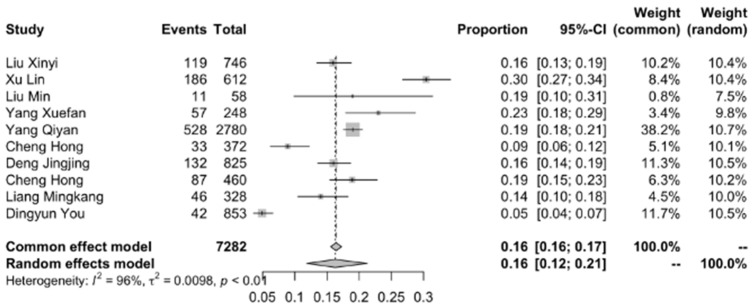
Forest plot of meta-analysis among 12 studies; random-effects model. Yunyou, D., et al. [13], Xinyi, L., et al. [14], Lin, X., et al. [15], Min, L., et al. [16], Jingjing, D., et al. [17].

**Table 1 ijerph-19-10207-t001:** Study characteristics.

First Author	Year of Publication	Study Location	Region	City Level	Type of Sample	Population Size	Method	Age	Sex	Vaccination (n)	Intention (n)	Completed (n)	Vaccination Intention Rate	Educational Level of Their Parents—Junior High School or Below (n)	Educational Level of Their Parents—High school or Vocational School (n)	Educational Level of Their Parents—Bachelor or above (n)	Other Important Information	Risk of Bias	Quality Grading
Dingyun You	2021	Kunming	Western China	second-tier	Mixed sample	1042	Injection	17–24	Female	42	853	147	0.819	336	409	297	Sufficient knowledge acquisition for individuals is an essential component of immuzation process, so it is of great importance to compare the background of the participants [13].	0	3
Liu Xinyi	2021	Shijiazhuang	Northern China	second-tier	Mixed sample	1622	Injection	17–24	Female	119	746	89	0.4662	234	756	631	College students who had been vaccinated had already had sex and perceived the risk of infection more consciously [14].	0	5
Xu Lin	2021	Guangzhou	Southern China	first-tier	Medical students	765	Injection	17–24	Female	186	612	175	0.881	124	254	387	Medical school offers courses on HPV [15].	0	4
Liu Min	2021	Weifang	Northern China	third-tier	Mixed sample	408	Injection	17–24	Female	11	58	9	0.6123	99	101	208	Among the samples, 567 (74.1%) were medical majors and 456 (59.6%) were from rural areas. The group of people vaccinated were senior medical students with knowledge of HPV [16].	0	4
Yang Xuefan	2021	Chongqing	Western China	first-tier	Mixed sample	566	Injection	17–24	Female	57	248	45	0.464	163	251	152	N/A *	0	5
Yang Qiyan	2021	Haikou	Southern China	second-tier	Mixed sample	3476	Injection	17–24	Female	528	2780	497	0.781	991	1254	1231	N/A *	0	3
Cheng Hong	2021	Fuyang	Southern China	third-tier	Mixed sample	620	Injection	17–24	Female	33	372	28	0.5968	132	204	284	N/A *	0	3
Deng Jingling	2021	Suzhou	Southern China	second-tier	Mixed sample	1353	Injection	17–24	Female	132	825	127	0.614	457	588	308	Female college students with a low vaccination cost and low self-efficacy were more likely to avoid vaccination [17].	0	4
Zhang Qiqi	2021	Wulumuqi	Northwest China	second-tier	Mixed sample	590	Injection	17–24	Female	87	460	68	0.7841	157	168	265	N/A *	0	3
Liang Minkang	2021	Lasa	Northwest China	second-tier	Tibetan students	800	Injection	17–24	Female	46	328	41	0.4113	459	257	84	Tibetan students are less willing to be vaccinated than Han students because of certain elements of their religion [18].	1	3
Fang Lan	2021	Nanning	Southern China	second-tier	Mixed sample	350	Injection	17–24	Female	41	198	33	0.568	67	154	124	N/A *	0	4
Cheng Ranran	2021	Handan	Northern China	third-tier	Mixed sample	1008	Injection	17–24	Female	39	795	31	0.789	323	451	234	N/A *	0	4

Cities are rated according to the 2021 City Business Charm Ranking [19]. N/A * shown irrelevance of key findings.

**Table 2 ijerph-19-10207-t002:** Subgroup analysis.

Subgroup	Sample Size	Heterogeneity Test Result (%): I^2^	Heterogeneity Test Result *p*-Value	Effect Model	Integrated Intention Rate (95% CI)	*Z*-Value	*p*-Value
City level: first-tier-city	1331	94.92	<0.001	Random-effects model	17.64% (95% CI: 12.76–21.63%)	0.1539	<0.001
City level: second-tier-city	9022	96.65	<0.001	Random-effects model	15.39% (95% CI: 11.74–19.82%)	−1.1539	<0.001
City level: third-tier-city	2036	95.5	<0.001	Random-effects model	13.87% (95% CI: 9.36–15.65%)	−3.5474	<0.001
Type of participants: female non-medical college students	9993	96.47	<0.001	Random-effects model	15.53% (95% CI: 11.20–20.22%)	1.1459	<0.001
Type of participants: female medical college students	765	92.35	<0.001	Random-effects model	30.37% (95% CI: 28.88–34.12%)	1.2156	<0.001
Type of participants: female Tibetan college students	800	92.73	<0.001	Random-effects model	14.12% (95% CI: 10.59%-18.04%)	−0.9872	<0.001
Educational level of their parents: junior high school or below (n)	3206	96.54	<0.001	Random-effects model	15.36% (95% CI: 11.59–17.54%)	0.083	<0.001
Educational level of their parents: high school or vocational school (n)	4438	94.28	<0.001	Random-effects model	17.18% (95% CI: 12.33–19.61%)	0.01717	<0.001
Educational level of their parents: bachelor or above (n)	3908	94.63	<0.001	Random-effects model	19.81% (95% CI: 15.61–22.25%)	−0.3068	<0.001

**Table 3 ijerph-19-10207-t003:** Results of the sensitivity analysis.

Eliminate the Literature	Vaccine Intention Rate	95% CI
Liu Xinyi	39.55%	0.36~0.63
Xu Lin	77.15%	0.45~079
Liu Min	51.38%	0.45~0.62
Yang Xuefan	56.46%	0.45~0.59
Yang Qiyan	69.16%	0.46~0.71
Cheng Hong	59.68%	0.45~0.62
Deng Jingjing	74.40%	0.45~0.79
Zhang Qiqi	69.16%	0.45~0.81
Liang Mingkang	37.59%	0.36~0.63
Feng Lan	66.48%	0.43~0.69
Cheng Ranran	80.87%	0.45~0.81
Ding Yuanyou	38.64%	0.35~0.67

## Data Availability

The data presented in this study are available on request from the corresponding author. The data are not publicly available due to privacy concern.

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
