# Peer review of "HPV Vaccination Intentions of Female Students in Chinese Universities: A Systematic Literature Review and Meta-Analysis"

_ijerph, 2022, doi:10.3390/ijerph191610207_

Round 1

Reviewer 1 Report

The study aimed to determine the HPV vaccination intentions of female college students in China

In my opinion abstract needs to be rewritten because it is too long. It is mainly the problem in the case of methodology and results. In addition, minor editorial errors, such as missing a period after a completed sentence, are present.

In the search strategy, I found keywords to be insufficient in terms of synonymous. There is no information about the use of MESH (via PubMed) or Emtree (via Embase), which could lead to not finding all relevant publications.

PRISMA flow diagram lack information regarding the reason for excluding the studies (incorrect methodology, intervention and population numbers (n)).

The description of the columns in table 1 is unreadable.

The description of figure 2 lacks an actual description.

Subgroup analysis:

Repetition of the words “medical students” is present in line 20.

Spaces are missing in several places – lines 17, 20, 22, 23

Table 4:

The description of the column should only include “Eliminate the literature”, and all the above information should be the description provided after point 3.2. before table 4.

Description of the abbreviation RMB should be introduced.

Conclusion

The sentence” …low intention to vaccinate HPV compared to other countries” should have some references or at least some examples.

Author Response

I am very honored to have your review and thank you for your patient and effective suggestions, which have helped me a lot to improve my manuscript. I have followed all of your suggestions one by one, here is the feedback, please refer to the newly uploaded manuscript for the specific changes.

The study aimed to determine the HPV vaccination intentions of female college students in China

In my opinion abstract needs to be rewritten because it is too long. It is mainly the problem in the case of methodology and results. In addition, minor editorial errors, such as missing a period after a completed sentence, are present.

The methodology part has been shortening.

In the search strategy, I found keywords to be insufficient in terms of synonymous. There is no information about the use of MESH (via PubMed) or Emtree (via Embase), which could lead to not finding all relevant publications.

After searching with more synonymous keywords including  ‘Vaccine ‘vaccination ‘HPV” “human papillomavirus’'immunization' ’injection’ ’acceptance’ ’ intention’;one more article falls in our inclusion criteria and we re-do the meta-analysis.

PRISMA flow diagram lack information regarding the reason for excluding the studies (incorrect methodology, intervention and population numbers (n)).

The flow gram has been re-writen.

The description of the columns in table 1 is unreadable.

Adjust the font and table displayed format.

The description of figure 2 lacks an actual description.

Added in footnote.

Subgroup analysis:

Repetition of the words “medical students” is present in line 20.

Canceled.

Spaces are missing in several places – lines 17, 20, 22, 23

fixed.

Table 4:

The description of the column should only include “Eliminate the literature”, and all the above information should be the description provided after point 3.2. before table 4.

the sentences have been moved out.

Description of the abbreviation RMB should be introduced.

added.

Conclusion

The sentence” …low intention to vaccinate HPV compared to other countries” should have some references or at least some examples.

example of Australia has been added.

Reviewer 2 Report

Thank you that you give me opportunity to review this manuscript  HPV Vaccination Intention of Female Students in Chinese Universities: A Systematic Literature Review and Meta analysis”.

The authors presented a systematic review about approaches to HPV vaccination among female students in Chinese universities. An interesting and actual topic.

Some remarks:

-        Abstract

  Literature was searched from inception until October 1, 2021. Please enter exact from date.... to

-   Introduction

There is no citation order, please correct it throughout the manuscript.

Example:  3, 30, 4

The highest risk group for HPV infection are women between the ages of 19 and 44 years [3]. The Global Advisory Committee on Vaccine Safety (GACVS) has evaluated the safety of HPV vaccines on multiple occasions and concluded in 2017 that HPV vaccines have an extremely favorable safety profile [30]. The cumulative benefits of HPV vaccination far outweigh the risks, with the greatest benefit occurring when the entire course of HPV vaccination is completed prior to initiating sexual intercourse. The World Health Organization (WHO) recommends that the HPV vaccine be included in the national vaccine plan. Vaccinating female university students against HPV has been shown to be cost-effective[4]

In the introduction there are only the references: 1,2,3,4, 30, 26, 20,21, and where are the others ??

-        Results

There is no references numbering in the table 1.

-        Discussion

There is only 1 reference (32) in the discussion, and where are other references?

-        Besides, many times there are whole paragraphs without references, e.g.

Many scholars have investigated the level of awareness and vaccination coverage of HPV vaccine among Chinese women, and found that the awareness of HPV vaccine is generally insufficient and the number of vaccination is low in China.

Or another example:

The entire long paragraph is not properly referenced. References 20,21 do not refer to Chinese women.

The HPV bivalent vaccine obtained the first marketing approval in China in July 2016. The quadrivalent vaccine was launched in China in November 2016, and the nine-valent HPV vaccine was launched in China on April 28, 2018. Thus, all HPV vaccines are now available in China. The HPV vaccine is import dependent and is in high demand in China. The availability of bivalent and quadrivalent vaccines is generally covered, but the nine-valent vaccine is still in short supply. It is worth noting that China's local HPV re-search and development has been initiated, and eight institutions have been set up to work on HPV vaccine development, among which, Yunnan Watson's nine-valent vaccine has been approved for clinical research. The launch of the domestic HPV vaccine will alleviate the problem of insufficient HPV vaccine supply. In China, the bivalent vaccine is available for women aged 9-45 years. The quadrivalent vaccine is available for ages 20-45, and the nine-valentvaccine is available for ages 16-26. The absence of a national immunization program and the age limit for vaccination determine the difficult environment for HPV vaccination in China, and there is a large gap with developed regions. Many scholars have investigated the level of awareness and vaccination coverage of HPV vaccine among Chinese women, and found that the awareness of HPV vaccine is generally insufficient and the number of vaccination is low in China. The literature reported that negative attitude and behaviours of Chinese female college student in the mainland such as lack of knowledge and low awareness about mode of transmission, protection, and early diagnosis methods of HPV infection, cost of HPV vaccine, potential adverse effects, and suspecting vaccine safety, and negative news on all vaccines prevented the generalisation of the HPV vaccination (20-21) The entire long paragraph is not properly referenced. References 20,21 do not refer to Chinese women.

The level of awareness and acceptance of HPV and HPV vaccine in Hong Kong, Macao and Taiwan is generally higher than that in the Mainland. Hong Kong female residents them-selves have high acceptance of HPV vaccine, but their acceptance of vaccination for their children is low, and there is a low-vaccination rate in practice. This systematic review included full-time female college students aged 17-24 years old who constitute a large portion of the vulnerable population aged 17 to 44. The vaccination is more effective at younger ages under the condition of vaccination availability, and it is more likely to be efficient in controlling the infection among the risk group of students who are sexually active if they can receive an effective transmission intervention. HPV vaccine is a class II vaccine with the characteristics of a commodity in China, college students have a certain self-determination in terms of their financial situation, and their willingness to receive HPV vaccination depends more on their own intention but also influenced by their parents' opinion toward the vaccine which directly affects the vaccination rate. There is no references!!

We included original cross-sectional and cohort studies that reported on HPV vaccination status of female college students in China (aged 17 to 24 years). But in abstract is  Papers of any design were included if they reported on young women aged 18-25 years (college age) etc

Search terms: Chinese search term: ("宫颈癌疫苗" OR "人乳头瘤病毒疫苗" OR "HPV疫苗" OR "6HPV疫苗" OR "9HPV疫苗") AND "接种" AND ("意愿" OR "态度" OR "意向"). English search term: ("HPV" OR “HPV-6" OR “HPV-9" OR "papillomavirus virus vaccine") AND ("attitude*" OR "opinion*" OR "intention*"). Please provide English words and no Chinese expressions

I understand  that宫颈癌疫苗 means Cervical Cancer Vaccine, but please change into English.

 Please correct the manuscript thoroughly.

Author Response

I am very honored to have your review and thank you for your patient and effective suggestions, which have helped me a lot to improve my manuscript. I have followed all of your suggestions one by one, here is the feedback, please refer to the newly uploaded manuscript for the specific changes. Your comment in black and my reply will be lighted in yellow for your reference.

Some remarks:

-        Abstract

  Literature was searched from inception until October 1, 2021. Please enter exact from date.... to

to shorten the abstract this information has been canceled.

-   Introduction

There is no citation order, please correct it throughout the manuscript.

the reference has been put into ordinal number.

In the introduction there are only the references: 1,2,3,4, 30, 26, 20,21, and where are the others ??

 added

-        Results

There is no references numbering in the table 1.

added in table 1

-        Discussion

There is only 1 reference (32) in the discussion, and where are other references?

added

-        Besides, many times there are whole paragraphs without references, e.g.

Meness of HPV vaccine is generally insufficient and the number of vaccination is low in China. The literature reported that negative attitude and behaviours of Chinese female college student in the mainland such as lack of knowledge and low awareness about mode of transmission, protection, and early diagnosis methods of HPV infection, cost of HPV vaccine, potential adverse effects, and suspecting vaccine safety, and negative news on all vaccines prevented the generalisation of the HPV vaccination (20-21) The entire long paragraph is not properly referenced. References 20,21 do not refer to Chinese women.

We included original cross-sectional and cohort studies that reported on HPV vaccination status of female college students in China (aged 17 to 24 years). But in abstract is  Papers of any design were included if they reported on young women aged 18-25 years (college age) etc

fixed.

Search terms: Chinese search term: ("宫颈癌疫苗" OR "人乳头瘤病毒疫苗" OR "HPV疫苗" OR "6价HPV疫苗" OR "9价HPV疫苗") AND "接种" AND ("意愿" OR "态度" OR "意向"). English search term: ("HPV" OR “HPV-6" OR “HPV-9" OR "papillomavirus virus vaccine") AND ("attitude*" OR "opinion*" OR "intention*"). Please provide English words and no Chinese expressions

I understand  that宫颈癌疫苗 means Cervical Cancer Vaccine, but please change into English.

translated into english.

Thank you!

Reviewer 3 Report

Journal: Vaccines (ISSN 1660-4601)

Manuscript ID: ijerph-1761717

Title: HPV Vaccination Intention of Female Students in Chinese Universities: A Systematic Literature Review and Meta analysis

 Comments to the authors:

The review paper conducted a meta-analysis of the research literature to explore HPV vaccination intentions of female college students in China. I recommend the publication of this article.

Can the authors please check for punctuation errors in the manuscript especially abstract.

Author Response

I am very honored to have your review and thank you for your patient and effective suggestions, which have helped me a lot to improve my manuscript. I have followed all of your suggestions one by one, here is the feedback, please refer to the newly uploaded manuscript for the specific changes.

Reviewer 4 Report

The paper entitled “HPV Vaccination Intention of Female Students in Chinese Universities: A Systematic Literature Review and Meta analysis” deals with an interesting topic. The objective to evaluate Human Papillomavirus (HPV) vaccination intention among young female to establish strategies for improving HPV vaccination coverage, is an important public health goal.

However, the paper needs some changes:

In the Abstract is written that were included papers if they reported on young women aged 18-25 years, while in the rest of the manuscript the age mentioned is 17-24 years. Please, resolve this inconsistency.

Still in the abstract:

- always write HPV with capital letters;

- add the periods at the end of the sentences;

- delete the word respectively into the parenthesis in the following period: “….and 14.02% (95% CI: 10.46%-18.01%, respectively), respectively…”; 

- eliminate the double parenthesis in the following sentence: “….17.19% (95 % CI: 12.31% to 19.62%) ) ….”.

In the paragraph “Results”:

- the flowchart in Figure 1 is not clear. The numbers in the boxes don’t clearly describe the screening process. For example, the 354 potentially-relevant documents that led to the selection of the 11 papers are not reported.

- Please, improve the layout of Table 1 to make it more readable.

- In “Meta-analysis results” eliminate the double parenthesis in the following sentence: “…. 16.64% (95%CI: 12.48%-21.27%))

- In “3.2. Sensitivity analysis and publication bias” move the following part out of Table 4: “After excluding the low quality paper, the effect size was merged again, and the results did not change significantly, as shown in Table 4. Begger's test showed minimum publication bias (Z=0.08, P=0.9379).

Author Response

I am very honored to have your review and thank you for your patient and effective suggestions, which have helped me a lot to improve my manuscript. I have followed all of your suggestions one by one, here is the feedback, please refer to the newly uploaded manuscript for the specific changes.Your comment is in black and ,my response was highlighted in yellow fro easier reading.

***However, the paper needs some changes:

In the Abstract is written that were included papers if they reported on young women aged 18-25 years, while in the rest of the manuscript the age mentioned is 17-24 years. Please, resolve this inconsistency.

changed to consistent.

Still in the abstract:

  • always write HPV with capital letters;
  • Sorry for misunderstanding,Should abbreviations be capitalized?
  • add the periods at the end of the sentences;
  • yes,added
  • delete the word respectively into the parenthesis in the following period: “….and 14.02% (95% CI: 10.46%-18.01%, respectively), respectively…”;
  • deleted
  • eliminate the double parenthesis in the following sentence: “….17.19% (95 % CI: 12.31% to 19.62%) ) ….”.
  • deleted

In the paragraph “Results”:

  • the flowchart in Figure 1 is not clear. The numbers in the boxes don’t clearly describe the screening process. For example, the 354 potentially-relevant documents that led to the selection of the 11 papers are not reported.
  • flow chart has been re-written.
  • Please, improve the layout of Table 1 to make it more readable.
  • the format and font has been changed
  • In “Meta-analysis results” eliminate the double parenthesis in the following sentence: “…. 16.64% (95%CI: 12.48%-21.27%))
  • deleted
  • In “3.2. Sensitivity analysis and publication bias” move the following part out of Table 4: “After excluding the low quality paper, the effect size was merged again, and the results did not change significantly, as shown in Table 4. Begger's test showed minimum publication bias (Z=0.08, P=0.9379).
  • moved the content

Round 2

Reviewer 1 Report

I am mostly satisfied with the adjustment made, but I have one additional concern related to the PRISMA diagram. I would rewrite it entirely because now it is misleading to false conclusions.
n=408 + 2 - this is the starting number of record
n=292 - removing duplication
n= 116 - then you need to read abstract and titles and exclude them
n=36 - full text read
n= number of exclusion and the reasons, you proved information n=32 while 7+6+4+1+2= 20. So it is hard to know how many publications were exluded finally.
n= studies included, which could be 12 (if 24 exclusion), 16 (if 20) or 4 (if 32).

This section must be rewritten.

Author Response

Thank you so much for your careful suggestions, this diagram is very important for the article and I have made the changes according to your comments.